### FLARE-GMM: an automatic aerosol typing model based on Mie-Raman-fluorescence lidar measurements with LILAS

Robin Miri<sup>1</sup>, Olivier Pujol<sup>1</sup>, Qiaoyun Hu<sup>1</sup>, Philippe Goloub<sup>1</sup>, Igor Veselovskii<sup>2,3</sup>, Thierry Podvin<sup>1</sup>, Fabrice Ducos<sup>1</sup>

**Abstract.** This study presents the development of an automated aerosol typing model utilizing Mie-Raman-fluorescence lidar

<sup>1</sup>Univ. Lille, CNRS, UMR 8518 – LOA – Laboratoire d'Optique Atmosphérique, Villeneuve d'Ascq 59650, France
 <sup>2</sup>Prokhorov General Physics Institute of the Russian Academy of Sciences, Moscow, Russia
 <sup>3</sup>Cimel Electronique, 172 rue de Charonne 75011 Paris, France

Correspondence to: Robin Miri (robin.miri@univ-lille.fr)

data collected by LILAS (Lille Lidar for Atmospheric Study), located on the ATOLL (ATmospheric Observations at LiLLe) platform in Lille, France. The proposed model, FLARE-GMM (Fluorescence Lidar based Aerosol REcognition from Gaussian Mixture Model), employs a Gaussian Mixture Model trained on a dataset spanning from early 2021 to the end of 2023. FLARE-GMM is able to distinguish the predominant aerosol type in a given layer between dust, urban and biomass burning aerosols by using the *PLDR* (Particular linear depolarization ratio) and the fluorescence capacity as well as RH, all measured with LILAS.
 To ensure accurate model training, cases were manually selected to include only pure aerosol layers, as mixed aerosols are not accurately modelled by GMM. Following the training phase, the model's performance was evaluated by investigating extreme events in which the aerosol type is not ambiguous. This approach was also completed with the use of a test dataset on which FLARE-GMM was compared to NATALI (Neural Network Aerosol Typing Algorithm Based on Lidar Data), another automatic aerosol typing model based on neural networks using lidar data. The results demonstrated that FLARE-GMM shows promise in accurately identifying aerosol types, indicating its potential for classifying aerosols in a variety of situations. Finally, FLARE-GMM was used to estimate the aerosol types present in Lille's atmosphere throughout the entire dataset from early

2021 to the end of 2023. A statistical analysis of these results was conducted, further underscoring the model's capability in

#### **Key words:**

automated aerosol classification.

Aerosol typing, Fluorescence lidar, clustering, Gaussian Mixture Model

#### 1 Introduction

Aerosols are critical components of the atmosphere, significantly influencing the Earth climate system. They are emitted both by natural sources, such as pollen or marine aerosols, and anthropologic sources, like traffic or fossil fuel burning. In addition to

their effects on health, they interact with radiation, directly affecting the Earth radiative budget (direct effect). Their presence also disrupts the water cycle, affecting cloud properties and further modifying the Earth radiative balance through indirect and semi-direct effects (Twomey, 1959; Johnson et al., 2004; Seinfeld et al., 2016; Thorsen et al., 2020; Herbert et al., 2020). Compared to other climate forcers, aerosols have much shorter lifetimes in the atmosphere. Consequently, their distribution across the globe is highly heterogeneous and heavily dependent on their sources as well as the atmospheric dynamics (Inness et al., 2019; Intergovernmental Panel on Climate Change (IPCC), 2023).




Therefore, observations are essential to monitor their presence around the globe. Spaceborne instruments allow us to cover global scale at the cost of strong technical constraints and high expenses. They are complemented by ground-based systems which are more versatile and cheaper to develop, but that are limited to local measurements. To compensate this drawback, ground based instruments work with networks such as AERONET (AErosol RObotic NETwork), established by NASA (National Aeronautics and Space Administration) and PHOTON (PHOtométrie pour le Traitement Opérationnel de Normalisation Satellitaire), which gathers ground-based observations to retrieve aerosol properties around the globe. Among ground-based instruments, in situ systems like the Scanning Mobility Particle Sizer (SMPS), Aerosol Chemical Speciation Monitor (ACSM), and nephelometers can directly measure particle microphysical, chemical, and optical properties, enabling accurate characterization of aerosols (Anderson and Ogren, 1998; Ng et al., 2011; Coquelin et al., 2013). However, these instruments are limited to measurements at their specific locations, restricting their ability to characterize the entire atmosphere. To address this limitation, remote sensing instruments such as photometers and lidars are used. Photometers measure integrated aerosol optical properties over the atmospheric column, while lidars provide profiles of aerosol optical properties throughout the atmosphere. From such measurements, non-analytical inversion models and processes, such as MLE (Maximum Likelihood Estimation) or EB (Empirical Bayes) approaches, enable the determination of the microphysical and chemical properties of the particles (Warren and Vanderbeek, 2007; Chang et al., 2022). Lately, with the advancement of these instruments and the automation of measurement processes, the volume of data has increased exponentially. This surge in data availability has facilitated the development of data-driven inversion approaches, such as machine learning and deep learning, for the retrieval of aerosol properties from remote sensing measurements (Nicolae et al., 2018; Lolli, 2023).

LILAS (Lille Lidar for Atmospheric Study) is a Mie-Raman lidar located at the ATOLL (ATmospheric Observations at LiLLe) platform in Lille (France), managed by the Laboratoire d'Optique Atmosphérique (LOA), and is employed in the frame of EARLINET/ACTRIS-FR (European Aerosol Research Lidar Network/Aerosols, Clouds and Trace Gases Research Infrastructure-France). The unique feature of this instrument is its ability to measure atmospheric fluorescence induced by the laser pulse emission. This type of measurement is still relatively novel and is found in only a few atmospheric lidars (Rao et al., 2018; Reichardt et al., 2022; Gast et al., 2024). However, it shows great promise due to its high sensitivity to the bio-molecules, like chlorophyll, contained in some aerosols. This sensitivity enables for the discrimination between aerosols with high biogenic content, such as pollens and biomass burning smoke, from those with low biogenic content, such as desert dust or urban aerosols, allowing us to perform aerosol typing (Immler et al., 2005; Sugimoto et al., 2012; Veselovskii et al., 2020).

Another particularity of LILAS is the large volume of data it generates thanks to its high level of automation, enabling for nearly continuous operation. Additionally, it has been measuring fluorescence signals by night since 2019 and has been simultaneously measuring water vapor by night since early 2021, resulting in a substantial dataset spanning from the early of 2021 to the end of 2023 of night-time LILAS measurements available for this study.

The growing amount of data acquired by LILAS, combined with other limitations which will be discussed later, motivated us to work on an automatic aerosol typing method in order to improve the manual approach described in Veselovskii et al. (2022). The objective of this work is to use the specificities of the LILAS instrument to train a new machine learning algorithm we have called FLARE-GMM (Fluorescence Lidar Aerosol REcognition with Gaussian Mixture Model). Its objective is to automatically perform aerosol typing out of LILAS measurements. The first part of this paper presents the instrument and FLARE-GMM, highlighting its distinctive features and advantages compared to other aerosol typing models. The second part details the training phase of FLARE-GMM, describing the methods used to select the model hyperparameters and to assemble the training set. The third part evaluates the model performance by analyzing extreme events and comparing its results with those of NATALI (Neural Network Aerosol Typing Algorithm Based on Lidar Data), another automatic aerosol typing algorithm using lidar data. Finally, before concluding and proposing future directions, a statistical study of the aerosol types present in Lille atmosphere obtained with FLARE-GMM is presented.

#### 2. Instrument and model presentation

#### 2.1 LILAS instrument and data

LILAS is a Mie-Raman lidar located at the ATOLL platform in Lille, France (50.611° N, 3.138° E). It's emission part consists in a Nd:YAG, doubled and tripled in frequency, operating at a repetition rate of 20 Hz with a pulse energy of 100 mJ at 355 nm. The lidar system is configured in a  $3\beta + 2\alpha + 3\delta$  arrangement. This setup allows us to retrieve the elastic backscatter coefficients and the particle linear depolarization ratios of aerosols at the wavelengths of emission (1064 nm, 532 nm, and 355 nm). Additionally, it measures the extinction coefficients of aerosols at 532 nm and 355 nm. The instrument also includes a Raman channel centered at 408 nm for monitoring atmospheric humidity and a detection channel ranging from 444 nm to 488 nm devoted to laser-induced atmospheric fluorescence observation. As already mentioned, this instrument is highly automated and therefore operates almost continuously when it is not raining.

The gathered lidar profiles are inverted using the modified Raman inversion method (Ansmann et al., 1992; Veselovskii et al., 2022) used to obtain quicklooks of the elastic backscatter coefficient ( $\beta_{\lambda}$ ) and the Particular Linear Depolarization Ratio (PLDR) of the aerosols at 532 nm, as well as the fluorescence backscatter coefficient ( $\beta_{fluo}$ ) of the aerosols, and the water vapor mixing ratio, with a high temporal and vertical resolution (about 3 minutes and 7.5 m respectively). In order to maximize the signal to noise ratio, the quicklooks are averaged over a period of 1 hour. Only nighttime measurements have been


considered for this study because of the low intensity of fluorescence and water vapor Raman signals, which makes it difficult obtaining accurate daytime measurement.

In order to calibrate the water vapor mixing ratio measured by the lidar, the method developed by Foth et al. (2015) is used with a RPG-HATPRO microwave radiometer and temperature data obtained from the ERA-5 reanalysis model. The lidar water vapor mixing ratio calibration procedure is described in Miri et al. (2024). From this calibrated measurement and with the ERA-5 temperature, the atmospheric Relative Humidity (RH) is computed.

More details about the LILAS instrument and data management can be found in Hu et al. (2018) and Veselovskii et al. (2020).

| Quantity         | $oldsymbol{eta}_{532}$ | $PLDR_{532}$ | $oldsymbol{eta}_{	extit{fluo}}$ | RH    |  |  |  |  |
|------------------|------------------------|--------------|---------------------------------|-------|--|--|--|--|
| Units            | $sr^{-1}.m^{-1}$       | Ø            | $sr^{-1}.m^{-1}$                | ∅,%   |  |  |  |  |
| Availability     | All                    | All          | Night                           | Night |  |  |  |  |
| Time Resolution  | 3 min                  |              |                                 |       |  |  |  |  |
| (initial)        |                        |              |                                 |       |  |  |  |  |
| Time Resolution  | 1 H                    |              |                                 |       |  |  |  |  |
| (final)          |                        |              |                                 |       |  |  |  |  |
| Range Resolution | 7.5 m                  |              |                                 |       |  |  |  |  |
|                  |                        |              |                                 |       |  |  |  |  |

Table 1: LILAS summary table of the various important quantities used in this study with their characteristics  $\beta_{532}$  is the elastic backscatter coefficient of the aerosols at 532 nm,  $PLD\,R_{532}$  the PLDR of the aerosols at 532 nm, and  $\beta_{fluo}$  the fluorescence backscatter of the aerosols.

#### 2.2 Choice of the model


As explained in the introduction, the objective of this study is to develop a machine learning model able to automatically identify the aerosol type from LILAS measurement and which exploits the instrument ability to measure atmospheric fluorescence. In Veselovskii et al. (2022), an approach is proposed to distinguish between dust, urban, smoke and pollen aerosols using LILAS measurement. It is based on the aerosol PLDR at 532 nm, and the fluorescence capacity ( $G_{fluo}$ ), which is defined as the ratio  $\beta_{fluo}/\beta_{532}$ , to estimate the aerosol type, according to the classification represented Figure 1.

Figure 1: Aerosol type with a depolarization/fluorescence capacity diagram. (adapted from Veselovskii et al. (2022)).

Based on this distribution, the Fluorescence Lidar based Aerosol REcognition from Gaussian Mixture Model (FLARE-GMM) algorithm has been developed to automatically estimate the predominant aerosol type of a given layer from LILAS measurements using a data-driven machine learning approach. FLARE-GMM leverages a Gaussian Mixture Model (GMM), a probabilistic clustering method that models the data as a combination of several Gaussian distributions, to identify patterns in the datasets and facilitate robust classification. The algorithm considers the *PLDR* at 532 nm, RH, and the fluorescence capacity, to identify the aerosol type.

One major advantage of machine learning is that it learns the decision boundaries from the data with statistical processes, thereby reducing reliance on manual thresholding and simplifying the analysis of complex, high-dimensional feature spaces.

A notable challenge in aerosol classification, as highlighted by Veselovskii et al. (2022), is the influence of hygroscopic growth. This phenomenon, in which aerosol particles interact with moisture, consequently alters their optical properties as a function of relative humidity, can significantly compromise the accuracy of aerosol typing. By incorporating RH as an additional feature in the model, FLARE-GMM effectively accounts for the influence of hygroscopic growth, thus enhancing the reliability of the aerosol retrieval process under varying environmental conditions.

Then, compared to other machine learning algorithms, GMM shows many benefits. Compared to K-means, a simpler clustering algorithm, GMM algorithms provide more detailed information and enable finer classification, at the cost of longer computation time. K-means is a non-probabilistic model that uses a hard-clustering approach, meaning that it gives a binary information about aerosol type, whereas GMMs are probabilistic models. This distinction allows GMMs to handle cluster overlap more efficiently than K-means, as well as better dealing with uncertainty, since clusters are represented by Gaussian distributions. This representation allows for better identification of outliers and data points near decision boundaries, a



capability that K-means lacks since it only indicates the class to which each data point belongs (Bishop, 2006; Patel and Kushwaha, 2020).

Eventually, compared to neural network methods, GMM also has many advantages. Neural networks are popular algorithms, allowing to solve very complex problems, and have been used in various occasions for aerosol typing (Nicolae et al., 2018; Papagiannopoulos et al., 2018; Voudouri et al., 2019). The ability of neural networks to solve complex problems lies in the number of their parameters, which can be important depending on the network complexity. However, the high number of parameters means that the computation time for training neural networks can be significant. Additionally, a very large training set is necessary to prevent overfitting, a strong limitation of these algorithms. Finally, neural networks are generally supervised models, implying that the training set already contains the expected aerosol type. Constituting the training set can therefore represent an important challenge if the classification of aerosols is performed manually, moreover considering that the training set size needs to be large in order to correctly train the neural network. This aspect has motivated researchers to work with simulated data, as it is the case for Nicolae et al. (2018), in order to work with a large training set containing the aerosol type of each data point. The advantage of GMM compared to neural networks lies in the ability to work with real, unclassified data, whereas neural networks require simulated labelled data for training.

Nonetheless, the primary limitation of GMM is its foundational assumption that the data are generated by Gaussian processes.

When the actual data distribution deviates from this assumption, accurately identifying clusters becomes problematic. Lidar instruments (such as LILAS) often measure an averaged optical property over a volume containing mixtures of different aerosol types. In these cases, the recorded optical properties represent a convolution of the individual contributions from various aerosol sources. This convolution effect is not well modelled by GMM during the training phase, which may lead to biases in cluster determination, and is a huge challenge in the selection of the training set to correctly identify the different clusters.

#### 3. FLARE-GMM training

#### 3.1 Data preparation





To correctly train GMM algorithms, the training set, composed of LILAS measurements from early 2021 to the end of 2023, needs to be pre-processed. Before rescaling the dataset to make sure that each variable contributes equally to the cluster identification, it needs to be filtered, as the presence of outliers in the training set can strongly impact the model.

In order to filter LILAS data, the PLDR at 532 nm,  $\beta_{532}$ ,  $G_{fluo}$ , RH and the altitude have been considered. First concerning the PLDR, situations in which the measured depolarization ratio is negative have been filtered out since negative values of PLDR are non-physical. It means that a problem occurred during the profile inversion. This concerns about 5% of the dataset. Then, regarding the upper limit, we have chosen to fix it at 40%. This choice has been motivated by the maximum values reached by desert dust aerosols (Haarig et al., 2022). Pollen aerosols may exhibit higher PLDR values at 532 nm, up to 80%

(Bohlmann et al., 2021), but, as it will be shown later, occurrences of pure pollen aerosols have not been observed in the dataset. Instead, pollen is often mixed with urban aerosols in the atmospheric boundary layer, making it difficult to reach such levels of depolarization. Situations for which the PLDR value reaches higher values than 40% may correspond then to ice clouds, which can be misclassified as desert dust aerosols otherwise.

- Then,  $\beta_{532}$  has also been used to filter the dataset. The objective here is to filter out cloudy cases which can alter the data, either by impacting the profile inversion, or due to the screen effect of optically thick clouds. Therefore, after some tests, we have chosen to filter out profiles for which  $\beta_{532}$  reaches over  $10\,M\,m^{-1}\,s\,r^{-1}$  to avoid cloudy situations. Moreover,  $\beta_{532}$  has also been used to filter situations with low aerosol load. It is important to remove these situations as performing aerosol typing if there is little to no aerosol is not relevant. But moreover, if  $\beta_{532}$  is low, the PLDR becomes very sensitive to measurement noise. Indeed, the depolarization is computed from a ratio between  $\beta_{532}$  measured in parallel and cross polarization states. Therefore, as these values decrease under low aerosol load conditions, the PLDR sensitivity to measurement noise increases and can reach outlier values. Hence, if  $\beta_{532} < 0.7\,M\,m^{-1}\,s\,r^{-1}$ , the case is filtered out from the training set. These upper and lower thresholds have also been applied to the results of FLARE-GMM, and these situations have been classified as "clouds" and "background", respectively.
- Regarding  $G_{fluo}$ , the results from Veselovskii et al. (2022) and the observation of the LILAS data have been used to determine the various thresholds to filter outliers. In the dataset used for this study, smoke aerosols can have  $G_{fluo}$  reaching up to  $10^{-3}$ . This value has then been chosen as the upper threshold for  $G_{fluo}$ , and cases exhibiting higher values have been filtered out. For the lower threshold, a value of  $9.10^{-6}$  has been selected. This choice has been motivated by observations of  $G_{fluo}$  of urban and dust cases, which rarely fall below this threshold. Lower values of  $G_{fluo}$  are typically observed only in clouds. This lower threshold is then a secondary filtering process enabling to exclude such conditions.
  - Then, concerning RH, negative values or values above 100% have been removed. This concerns about 0.5% of the dataset. And finally, only altitudes above 1000 m above ground level have been considered to ensure that LILAS overlap function is equal to 1, guaranteeing inversion quality. Similarly, only altitudes below 6000 m have been considered in the first place to maintain an acceptable signal-to-noise ratio for the RH data.
- A next step is to pre-process the data. The fluorescence capacity can vary on very large ranges (between  $9.10^{-6}$  to  $10^{-3}$ ), compared to PLDR and RH, which typically vary within a single order of magnitude. This wide variation implies that some information may be lost during the rescaling process. To mitigate this effect, the base-10 logarithm of  $G_{fluo}$  has been used as a feature for the GMM training. In this way, the range of  $G_{fluo}$  is squeezed, reducing the disparity with other variables. Afterward, the training set has been rescaled by retracting the mean of the dataset and dividing by the variance, to ensure that each variable

contributes equally to cluster identification. This rescaling step is crucial as it balances the influence of each variable in GMM, allowing the model to more effectively identify clusters without being dominated by any single variable.

| Quantity             | $oldsymbol{eta}_{532}$                             | PLDR | $G_{\mathit{fluo}}$ | RH   | Altitude |
|----------------------|----------------------------------------------------|------|---------------------|------|----------|
| Maximum<br>threshold | $10  \text{s}  r^{-1} M  m^{-1}$ (profile removed) | 40%  | 10 <sup>-3</sup>    | 100% | 6 km     |
| Minimum<br>threshold | $0.7\mathrm{sr}^{-1}M\mathrm{m}^{-1}$              | 0%   | $9.10^{-6}$         | 0%   | 1 km     |

Table 2: Summary of the various thresholds used to filter the dataset in the first time

#### 3.2 Training set construction






During the development of FLARE-GMM, multiple training datasets have been evaluated. In its current configuration, the training dataset comprises manually selected cases that are categorized into three subsets based on ambient humidity conditions. This categorization was implemented to address specific challenges encountered during the training.

The first issue encountered was the presence of aerosol layers composed of mixed aerosol types. As already discussed, such mixtures are not well represented by GMM algorithms, since the measurements obtained by LILAS reflect a convolution between the Gaussian distributions of each individual aerosol component. This phenomenon results in data points being located in the interstitial spaces between clusters, complicating the identification process during the training phase. An initial attempt to mitigate this issue involved excluding the atmospheric boundary layer from the training dataset, because this region typically contains aerosol layers with mixed types. However, this approach was unsuccessful. Consequently, a manual analysis of each case has been operated to selectively include only cases featuring aerosol layers that are likely to be made up of a single aerosol type rather than a mixture. This identification process relied both on the quadrant method described in Veselovskii et al. (2022) and shown in Figure 1, as well as an early version of FLARE-GMM, trained on the dataset excluding the boundary layer.

The second challenge encountered was the representation of hygroscopic growth. As noted earlier, it alters aerosol optical properties under humid conditions, complicating the classification. To address this, RH was initially incorporated as a feature in FLARE-GMM. However, the resulting cluster identification was not what was expected. This has been interpreted as due to the fact that hygroscopic growth is not well modelled by a Gaussian distribution. Specifically, for a given aerosol, there is a bijective relationship between its optical properties and RH. Consequently, the three-dimensional cluster in the [PLDR], fluorescence capacity, RH feature space adopts a cylindrical shape, since variations along the RH axis can be represented by a translation. This contradicts the GMM assumption that clusters follow a 3D Gaussian distribution. To mitigate this issue, the dataset has been partitioned into three subsets based on humidity levels and three distinct models have been trained: one for dry conditions, one for high-humidity conditions, and one for intermediate conditions. Dry data points have been selected for RH< 60%, very humid data points for RH> 80%, and data with 60% 

Figure 2: 2D histograms of the training set containing hand-selected data of pure cases as a function of RH levels (a) below 60 % (b) between 60 % and 80 %, (c) over 80%, (d) all data

The silhouette coefficient method (Dinh et al., 2019; Zhou and Gao, 2014), shown in the Appendix, has been used on each section of the dataset, to determine the ideal number to clusters. In each case, the obtained number is 3, in agreement with what we were expecting from the observation of the dataset 2D histograms.

FLARE-GMM models have been trained on each section of the training set. The results of the training set data repartition are displayed on Figure 3. We see that the different clusters are well defined and separated in each case. The association of each cluster to its aerosol type, urban, smoke and dust, is not ambiguous and can be performed straightforwardly.

Figure 3: Data repartition of the training set sections with the different versions of FLARE-GMM trained on these sections (a) section with RH < 60%, (b) 60% < RH < 80%, (c) RH > 80%, each colour is associated to a Gaussian distribution determined by FLARE-GMM

However, this distribution suffers from some limitations. First, it relies on a limited dataset and thus, the resulting model may not perform adequately on unseen data. For a robust generalisation, the training set should encompass all the potential scenarios that the model might encounter in practice. The manual selection process inherently restricts the diversity and size of the dataset, which can compromise the model ability to accurately classify new data.

The second issue concerns the management of mixed aerosol layers. If multiple aerosol types coexist within the same resolution volume, data points may lie near the boundaries of several clusters. Without proper handling, the automatic aerosol typing model might inadvertently assign these mixed cases to the most prevalent aerosol class simply because the data points are closest to its corresponding cluster.

In order to address these issues, the likelihood function can be used. It is defined as:

255 
$$P(x) = \sum_{j=1}^{K} \pi_j N(x|\mu_j, \sigma_j), \qquad (1)$$

Where  $\pi_j$  are the weights of each Gaussian and  $N\left(x\middle|\mu_j,\sigma_j\right)$  correspond the Gaussian distribution j (of mean  $\mu_j$  and standard deviation  $\sigma_j$ ) evaluated in the point x. This function can be interpreted as the probability that x has been generated by a Gaussian distribution of the model. Therefore, by choosing a threshold on the likelihood value, it is possible to balance between enlarging the clusters to mitigate the impact of the limited training set and excluding mixture cases and outliers from the repartition process. Indeed, if a low likelihood value is chosen as threshold, the resulting clusters will be narrower, enhancing the reliability of the aerosol typing estimation by FLARE-GMM, however in the same time, narrow clusters means that the model will have more difficulty to identify cases that were not in the training set and cases with higher measurement uncertainty. On the other hand, if a high likelihood value is chosen, the clusters will be wider, allowing to identify more cases that are potentially not contained in the training set, but potentially decreasing the reliability and the accuracy of the model.

Figure 4: Data repartition of the training set sections with the different versions of FLARE-GMM trained on these sections (a) section with RH < 60 %, (b) 60 % < RH < 80 %, (c) RH > 80 %, each color is associated to a Gaussian distribution determined by FLARE-GMM, with the representation of the negative log likelihood contour lines  $(-\ln(P(x)))$ 

Figure 4 shows the contour lines of the negative log likelihood field, i.e.  $-\ln(P(x))$ , as a function of the rescaled features.

This figure has been analysed in order to select the optimal threshold of  $-\ln(P(x))$ , and correctly filter out mixture cases and outliers, while also widening the clusters to consider the limits of the training set. Finally, the threshold has been fixed at 8.

25

This value has been chosen after running tests and allows for cases out of the training set, for which  $-\ln(P(x))$  is generally below 8, to be correctly classified, while also excluding mixture cases and outliers, for which  $-\ln(P(x))$  is generally over 8. Another benefit of this model is that it can be used at high altitudes, where RH is much more difficult to obtain accurately. In order to use FLARE-GMM above 6 km, altitude above which RH cannot be measured by LILAS, we have decided to use the driest model to classify cases. The reason is that lidar can measure fluorescence and depolarization at very high altitudes, and in clear sky conditions, the RH level being typically lower at these altitudes (Wolf et al., 2023). Consequently, hygroscopic growth cases are rarely detected at high altitudes. This encourages us to use a dry aerosol model for aerosol typing in such cases, allowing us to perform aerosol typing up to 15 km during the night.

Eventually, we see that the cluster associated with urban aerosols and the one associated with smoke particles are very close to each other. Moreover, there is no separation between them once the negative log likelihood criterion is applied. This makes it currently very difficult to differentiate layers containing a mixture of urban and smoke cases from a layer containing pure particles of one aerosol type. Instead, FLARE-GMM can estimate which aerosol type has the stronger contribution to the mixture optical properties. In our case, it is not possible to improve this result, as the clusters are too close. The use of other optical properties in future studies, such as the *LR* or the Ångström exponent, could help to solve this issue.

#### 4. Generalisation of FLARE-GMM

The objective of this section is to generalise FLARE-GMM. To do so, its results have been compared to aerosol type obtained from other methods for cases out of the training set. In the first part, it concerns cases of extreme events in which the aerosol type is not ambiguous and that has already been documented by other studies. In the second part, FLARE-GMM is compared to NATALI, another automatic aerosol typing model based on lidar data, which uses a neural network.

#### 4.1 Classification of specific events

290

Assessing the accuracy of clustering models such as GMM can be challenging in the absence of definitive reference. In this section, an analysis of specific events has been performed to have an idea of the algorithm performances to identify aerosol types from LILAS data.

Our initial approach involves exploring FLARE-GMM aerosol typing estimation in instances where aerosol types are not ambiguous and easily identified. These scenarios mainly manifest during specific events of dust or smoke occurrences. Fortunately, the region of Lille frequently experiences such events, which are consistently documented and analysed by the LOA, and which origins can be checked from backward trajectories (Baars et al., 2019; Draxler et al., 2023; Stein et al., 2015). The first event analysed in this section occurred during the night between 15 and 16 March 2022. In this period, strong manifestations of Sirocco winds were observed. They are responsible for the advection of Saharan desert dust over Europe. Consequently, desert dust can be observed in Lille during such events (Husar, 2004; Stohl et al., 2004). The backward trajectory

for this night (see Appendix) confirms that the air mass above Lille came from the Saharan region, thus supporting the fact that desert dust is expected to be observed in Lille.

Figure 5: FLARE-GMM aerosol type estimation quicklook during the night between 15 and 16 March 2022 between 1000 m and 15000 m

On the other hand, Figure 5 shows the quicklook of FLARE-GMM aerosol type estimate during the night between 15 and 16 March 2022. Background and cloud classes are automatically attributed when  $\beta_{532} < 0.5\,M\,m^{-1}\,s\,r^{-1}$  or  $\beta_{532} < 15\,M\,m^{-1}\,s\,r^{-1}$  respectively, while the unknown class gathers the outliers and mixture cases for which  $-\ln\left(P\left(x\right)\right) > 8$ . Figure 5 shows that after 00:00 UTC on 16 April 2022, an aerosol layer was present below 3000 m, with clouds at the top of the layer. FLARE-GMM estimates that these aerosols are certainly desert dust aerosols, which is consistent with the backward trajectories as well as the analyses and the different reports made on this particular situation (Bouteiller, 2022). FLARE-GMM aerosol typing estimate concurs with the expected result in this case, supporting the fact that FLARE-GMM is able to identify desert dust aerosols in such events. We can notice however that it is much more complicated for the aerosol layer above 6 km and between 20:00 and 01:00 UTC, which appears as a mixture between unknown aerosols, clouds and desert dust, but which are probably misclassified here. Furthermore, one can notice that from 1:00 UTC to the end of the night, a cloud is present at around 2.5 km, shadowing the layers above and making it impossible for the lidar to observe aerosol layers above it. The second case used in this study occurred during the night between 2 and 3 March 2021. Similarly to the previous case, strong Sirocco winds were responsible for the transport of Saharan desert dust over Europe. The backward trajectory for this case (see

320


of desert dust in the atmosphere. Figure 6 shows the quicklook of FLARE-GMM aerosol type estimate for this case. Here, FLARE-GMM identifies desert dust aerosols in a layer spanning from 2000 m to almost 7000 m. Clouds can also be observed after 00:00 UTC, ranging from 3000 m to 10000 m, while the lowest part of the atmosphere is associated with unknown aerosol type, which could correspond to a mixture between desert dust and urban aerosols in the boundary layer. Such as the previous studied case, in this situation, the ability of FLARE-GMM to correctly identify desert dust aerosol layers is illustrated here, as FLARE-GMM aerosol typing estimate corresponds to the aerosol type expected from backward trajectory and previous analyses (Veselovskii et al., 2022).




Figure 6: FLARE-GMM aerosol type estimation quicklook during the night between 2 and 3 March 2021 between 1000 m and 15000 m

The last case investigated in this part occurred during the night of 19 July 2022. Significant forest fires occurred in the Gascogne region in south-eastern of France. The winds blowing northward during this event transported biomass burning aerosols to Lille. This can be clearly observed in Figure 7 where both the fire map and the backward trajectory are represented. Figure 7 (a) shows the backward trajectory in this situation and Figure 7 (b) the fire map between 14 and 19 July 2022, obtained from the Fire Information for Resource Management System (FIRMS), which uses data from MODIS and the Visible Infrared Imaging Radiometer Suite (VIIRS), and is managed by NASA (source: https://firms.modaps.eosdis.nasa.gov/, last access: 28 June 2024). On this map is also highlighted the fire data corresponding to the forest fires that occurred in the Gascogne region at this period. Analysing both these figures, it is possible to observe that biomass burning aerosols emitted from the forest fires have been transported to Lille during this period. On the other hand, Figure 7 (c) shows the aerosol typing estimate from

FLARE-GMM during the night of 19 July 2022. This figure shows that FLARE-GMM correctly recognizes the presence of a smoke layer ranging from 2000 m to 6000 m with the presence of unknown aerosols in the lower part of the atmosphere, which could correspond to a mixture with another aerosol type like urbans, or to outliers. Nevertheless, this case illustrates well the ability of FLARE-GMM to identify smoke layers in such conditions, supporting its efficiency and generalisation.

Figure 7: (a) Backward trajectory of 24 hours at 4000 m above ground level at 22:00 UTC on 19 July 2022 (b) Fire map from the Fire Information for Resource Management System (FIRMS) between 14 to and 19 July 2022 (source: https://firms.modaps.eosdis.nasa.gov/, last access: 28 June 2024) (c) FLARE-GMM aerosol type estimation quick-look during the night of 19 July 2022 between 1000 m and 15000 m

These three presented cases allow for the evaluation of the performance of FLARE-GMM in occurrences of strong events. In these situations, the aerosol type estimated by the algorithm is consistent with the expected aerosol type observed in the atmosphere. The ability of FLARE-GMM to correctly identify aerosol types in such cases is thus supported by these examples. However, this approach is limited since it uses a low number of specific situations, which is therefore not ideal to evaluate the algorithm performance in general. In order to complete this approach, in the absence of absolute reference for the estimate of the aerosol type with lidar data, FLARE-GMM can be compared to another automatic aerosol typing method which uses lidar data.

#### 4.2 Comparison with NATALI aerosol typing

Neural Network Aerosol Typing Algorithm Based on Lidar Data (NATALI) is a deep learning algorithm developed to estimate the most probable aerosol type from lidar data. This algorithm uses the EARLINET  $3\beta + 2\alpha(+1\delta)$  profiles, which are multispectral profiles that can be obtained from LILAS and are regularly inverted. NATALI has been trained on synthetic data, using the aerosol Ångström exponent, colour index, colour ratios, LR and PLDR of the aerosols as features to perform aerosol typing. From these properties, the algorithm is able to determine aerosol type among continental, continental polluted, smoke, dust, marine and volcanic (Nicolae et al., 2018).

In this section, a comparison between NATALI and FLARE-GMM is presented so as to evaluate how both models perform compared to one another on LILAS data. 36 cloud-free profiles from 2022, covering different situations and aerosol types have been selected randomly to compare FLARE-GMM and NATALI estimates. Figure 8 shows the confusion matrix between the two aerosol type estimates. Confusion matrices are usually used to evaluate the performance of a classification algorithm. They display the counts of true positives, true negatives, false positives, and false negatives, helping to assess the accuracy, precision and overall models performance. In this case, this matrix can be used to compare the results from the two models, analyse their agreements and discrepancies.

The confusion matrix indicates that the agreement rate between the two models is at 38 %. This rate is encouraging given that the two models use different features for classification and differ in their algorithmic structure. Indeed, NATALI is a supervised learning algorithm, while FLARE-GMM is unsupervised. Regarding the disagreements between FLARE-GMM and NATALI, we can first evidence the confusion that exists between smoke and urban, or continental aerosols. Indeed, in a substantial number of cases, NATALI and FLARE-GMM disagree between smoke and urban aerosols. This confusion has been expected since the optical properties of smoke and urban aerosol are close, as evidenced in the former section.




Figure 8: Confusion matrix between FLARE-GMM and NATALI aerosol type estimation on 36 profiles from 2022




Moreover, we can notice that most cases identified as desert dust by FLARE-GMM are classified differently by NATALI, either as smoke (in 570 cases) or as marine aerosols (in 311 cases). This confusion is more surprising since desert dust optical properties are supposed to be different from these aerosol types. In particular, desert dust PLDR at 532 nm is expected to range close to 30 %, while smoke and marine aerosols are expected to exhibit much lower PLDR at this wavelength.

These differences between FLARE-GMM and NATALI can be explained by several factors. First, it could be a consequence of the difference between the two algorithms. As mentioned above, NATALI is a supervised learning model that has been trained on synthetic data, as opposed to FLARE-GMM that is an unsupervised learning model which has been trained using data from LILAS instrument specifically. This aspect is an advantage for FLARE-GMM as the specificities of the site in Lille, as well as the specificities of the instrument are therefore inherently integrated in the model. On the other hand, NATALI, which has been trained on synthetic data, might contain biases from the model used to simulate the aerosol optical properties. Moreover, the features used by NATALI might explain these differences with FLARE-GMM estimates. Indeed, NATALI uses the LR and the Ångström exponent, which both rely on extinction coefficients estimations. However, these properties are difficult to determine accurately with the Raman inversion, performed on the EARLINET profiles. These quantities often exhibit high uncertainties (Ansmann et al., 1992) thus impacting NATALI estimate quality. This might explain why NATALI predicts the presence of marine aerosols while the measurement site is located 70 km away from the coast, thus making it unlikely to observe aerosol layers mainly composed of marine particles. The case represented Figure 9 illustrates this situation. In this figure, both  $G_{fluo}$ 

and PLDR indicate that the aerosol layer is primarily composed of dust aerosols, characterized by a high depolarization ratio and low fluorescence capacity. FLARE-GMM correctly classifies most of the aerosol layer as dust. However, NATALI, which primarily relies on the Lidar Ratio (LR) and the Ångstrom exponent for aerosol type identification, classifies this aerosol layer as a combination of smoke and marine aerosols. This classification appears unlikely, particularly given the high depolarization ratio, suggesting that in this case, the Ångstrom exponent and Lidar Ratio may not be accurate enough for aerosol type estimate. This example highlights the challenges of using parameters like LR and Ångstrom exponent, which are highly sensitive to measurement noise, for aerosol classification, and shows that properties such as fluorescence capacity and depolarization ratio provide more reliable information to perform aerosol characterization.

Figure 9: LILAS lidar profiles on 21 March 2022 at 21:00 UTC and comparison between FLARE-GMM and NATALI aerosol types estimations

The treatment of hygroscopicity can also be responsible for the differences between NATALI and FLARE-GMM aerosol typing estimates. This phenomenon, which significantly alters aerosol properties, is taken into account by NATALI in the training set, as it is modelled according to different *RH* levels, but *RH* is not used as an input to determine the aerosol type (Nicolae et al., 2018). On the other hand, as FLARE-GMM uses real data in the training set, it covers a wide range of humidity levels, and furthermore, even if *RH* is not used as a feature, its influence on aerosol optical properties is considered with the use of different trained modeled in function of the *RH* levels. These differences of treatment can be responsible for important differences between the two estimates, especially between urban and smoke aerosols, as their optical properties can be difficult to distinguish if humidity is not considered.


Nevertheless, the comparison between FLARE-GMM and NATALI allows us to compare FLARE-GMM performance to another automatic aerosol typing model. While the comparison suffers from limitations that have been raised, making it challenging to formulate clear interpretations and conclusions, it is still providing encouraging results. Indeed, despite their differences, in terms of architecture, training methods and datasets used to perform the classification, the agreement rate between the two models is almost 40 %. Moreover, discrepancies between NATALI and FLARE-GMM can find explanations in many factors that have been mentioned. Therefore, this comparison, with the analysis of extreme events performed are promising for FLARE-GMM performances. They also provide a positive outlook for its potential future improvements and advancements, indicating that the model is robust and can continue to be improved.

#### 5. Aerosol type analysis in Lille

In this section, FLARE-GMM is used to analyse aerosol type estimates in Lille on all the available dataset. The advantage of developing an automatic aerosol typing method is that such analyses are easily performed quickly on a very large amount of data. The results can then be analysed to study aerosol properties in Lille, and evaluate potential trends.

By using FLARE-GMM on the LILAS dataset from 2021 to 2023, we can investigate the aerosol type repartition, as well as the seasonality of the aerosols in the Lille region. To do so, the aerosol type has been estimated by FLARE-GMM for each available profile between 2021 and 2023. Each profile containing more than 15 data points classified as a specific aerosol type have been considered in order to avoid treating outliers. Eventually, the considered altitudes for this analysis have been selected below 6 km. This choice has been motivated to avoid taking cirrus clouds into account, which might be classified as dust aerosols by FLARE-GMM due to their high depolarization and low fluorescence. Indeed, cirrus clouds often show low optical thickness and are therefore more complicated to differentiate from aerosols by using a criterion on the elastic backscatter coefficient. By considering only the data below 6 km, it is possible to mitigate the impact of cirrus clouds in the statistics while also considering most of the aerosol cases, which are in majority present in low altitudes.

Figure 10: (a) Violin plots and (b) Box plots of averaged altitude distributions, between 1 km and 6 km above ground level as a function of the aerosol type for all the available data from 2021 to 2023




First, it can be interesting to investigate the altitude distribution for each aerosol type. Figure 10 shows the violin plots (a) and the box plots (b) of the averaged altitudes above the ground of the identified aerosol layers as a function of the aerosol type. This plot indicates that both urban and smoke aerosols are predominantly detected at low altitudes, mainly within the boundary layer. The distribution for smoke aerosols exhibits a longer tail compared to urban aerosols. This is expected, as smoke aerosols, generally originating from fires, are emitted at high temperatures and can be injected into higher altitudes. In contrast, urban aerosols usually remain confined to the boundary layer and rarely reach higher altitudes. Regarding dust aerosols, their distribution shows that they can be present at much higher altitudes. This result can be interpreted in different ways. First, it could be a consequence of ice cloud detection, however, below 6 km, the presence of ice clouds is less probable in the Lille atmosphere. This could also be due to the fact that the primary sources of dust in Lille are not local. Unlike urban and smoke aerosols, which may be emitted locally, dust often originates from the Sahara or other deserts and is carried to Lille by the wind. As a result, dust particles are found at more dispersed altitudes compared to urban and smoke aerosols.

Figure 11 (a) shows the histogram of FLARE-GMM aerosol type estimates on all the available data below 6000 m as a function of time. This figure shows that in general, more aerosol layers are identified during spring and summer. This situation is influenced by Lille's meteorology, as it often rains in this region during winter. LILAS does not measure during rain, which reduces the amount of available data for analysis. Additionally, during winter, the boundary layer is generally lower due to decreased temperatures. Given that the minimum considered altitude is 1000 m, aerosol layers may not be detected, further limiting the data.

Figure 11: (a) Histogram of aerosol type estimate from FLARE-GMM as a function of the time and (b) share of each aerosol according to the season, with all data from 2021, 2022 and 2023 below 6000 m above ground level

Figure 11 (b) shows the seasonal share of each aerosol type estimated by FLARE-GMM. This figure first illustrates that urban aerosols are the main aerosol type in the Lille atmosphere, and represent more than half cases in each season. This result is expected as the LOA is close to the city of Lille, the emission of urban aerosols by human activity is therefore the first aerosol source in the observed atmosphere. Regarding smoke aerosols, Figure 11 (b) indicates that they are significantly more frequent during spring and summer compared to fall and winter. This trend can be attributed to higher temperatures in spring and summer, which increase the likelihood of fires, the primary source of smoke aerosols, occurring during these periods. Eventually, regarding dust aerosols, they are the least represented aerosol type. This is because dust scenarios are rare in Lille, which is not located close to a source of desert dust. The occurrence of dust cases in Lille are rather due to extreme events such as advection of Saharan dust by Sirocco winds as mentioned previously. Such events generally occur early in spring or in winter, and it is possible to observe that dust cases are more represented at these periods, thus confirming the importance of

these phenomena in the observation of dust aerosols in Lille. However, it is important to consider the proportion of dust cases in winter within the context of data availability limitations during this season. Additionally, the colder temperatures in winter increase the likelihood of observing ice or mixed-phase clouds below 6000 m. These clouds can occasionally be misinterpreted as desert dust aerosols with the current classification method, as it has been mentioned. Despite these challenges, these findings are crucial for gaining insights into the composition and distribution of aerosols in the region, showing the benefits from using an automatic aerosol typing process like FLARE-GMM.

#### Conclusion





In this study, we developed FLARE-GMM, a machine learning-based aerosol typing algorithm using lidar measurements from LILAS. By leveraging a Gaussian Mixture Model (GMM) trained on fluorescence capacity, depolarization ratio, and relative humidity, FLARE-GMM effectively classifies aerosol types between urban, dust and smoke, while addressing challenges such as hygroscopic growth and mixed aerosol layers.

A key advantage of FLARE-GMM lies in its ability to work with real, unclassified data, unlike supervised models that rely on synthetic training sets. Through a thorough evaluation using extreme aerosol events and comparison with the neural network-based NATALI algorithm, FLARE-GMM has demonstrated promising classification capabilities. However, certain limitations remain, notably in distinguishing urban and smoke aerosols in mixed layers and in the treatment of hygroscopic growth, but most importantly regarding the construction of the training set which has to be performed manually.

Applying FLARE-GMM to the full LILAS dataset from 2021 to 2023 provided valuable insights into the seasonal variability of aerosols over Lille, highlighting the dominance of urban aerosols and the episodic occurrence of smoke and dust events.

Due to the adaptable nature of this classification method, FLARE-GMM is well-suited to accommodate future technological advancements or algorithmic updates. For example, the new lidar system of LOA, LIFE (Laser Induced Fluorescence Explorer), operational since the end of 2024, will offer enhanced power and the ability to measure fluorescence across different wavelengths. This new capability is crucial for more accurate aerosol identification and will significantly deepen our understanding of aerosol types. By applying a protocol similar to the one detailed in this paper, it would be feasible to develop an updated version of FLARE-GMM that uses the LIFE dataset for training, further enhancing its capabilities and accuracy in aerosol typing. Finally, in order to assess the robustness of this approach to perform aerosol typing, it could be tested on another instrument also measuring aerosols but in a different environment than Lille, to confront its viability in the presence of other aerosol types such as marine, volcanic aerosols or pollen in higher quantity.

To effectively train future models using real data, it is essential to account for mixtures of different aerosol types. Instead of assuming that the training set is generated by a set of independent Gaussian distributions, an alternative approach would be to model data points as resulting from convolutions of multiple Gaussian distributions. This would enable the use of a significantly larger portion of the dataset, with only outliers being excluded. However, implementing such an approach would

necessitate more complex models with a larger number of parameters, increasing both computational demands and the complexity of the training process.

To conclude, this study underscores the potential of fluorescence lidar in aerosol classification and the benefits of unsupervised learning approaches for atmospheric studies. Future work will focus on improving aerosol mixture identification and incorporating additional optical parameters to refine classification accuracy.

#### Code and data availability

Data and code will be available upon the request

#### **Competing interests**

The contact author has declared that none of the authors has any competing interests.

#### Acknowledgements

We acknowledge CaPPA project funded by the ANR through the PIA under contract ANR-11-LABX-0005-01. The authors thank the Région Hauts-de-France, the Ministère de l'Enseignement Supérieur et de la Recherche and the European Fund for Regional Economic Development for their financial support to the CPER CLIMIBIO and ECRIN programs. The contribution from Q. Hu was supported by Agence Nationale de Recherche ANR (ANR-21-ESRE-0013) through the OBS4CLIM project. ChatGPT has been employed for the drafting purposes in this document.

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

#### Appendix

Figure A1: Silhouette coefficient from K-Means partitions for a number of clusters ranging from 3 to 8 on the sections of the training set (a) cases with RH < 60 %, (b) 60 % < RH < 80 %, (c) RH > 80 %

# NOAA HYSPLIT MODEL Backward trajectory ending at 0300 UTC 16 Mar 22 GDAS Meteorological Data

Figure A2: 100 hours backward trajectory at 1500 m above ground level at 03:00 UTC on 16 March 2022

## NOAA HYSPLIT MODEL Backward trajectory ending at 2200 UTC 02 Mar 21 GDAS Meteorological Data

Figure A3: 100 hours backward trajectory at 3000 m above ground level at 22:00 UTC on 2 March 2021