# Peer review of "FLARE-GMM: an automatic aerosol typing model based on Mie-Raman-fluorescence lidar measurements with LILAS"

_EGUsphere, 2025_

## Author Response (AR1)

**Reply to RC1**

**Specific comments:**

1) Line 220. The authors note that pollen aerosols are not clearly visible in the clusters, likely due to their low-altitude, local emission profile, below LILAS detection limits. Can the authors elaborate on how this limitation impacts the model's performance? Was there any attempt to identify mixed layers where pollen and urban aerosols coexist, even if pure pollen cannot be isolated?

Response: The low altitude emission of pollen has two consequences, the first one, as stated here is that aerosol layers containing pollen might fall below the detection limit of the instrument. The second one is that as they are emitted in the boundary layer, they get mixed with urban aerosols which are almost constantly present in the Lille atmosphere.

There were attempts to identify mixed layers containing pollen and urban aerosols. However, these layers most often exhibit depolarization ratio at 532 nm between 0.1 and 0.2, and fluorescence capacity between 0.8 10-4 and 1.2 10-4. The main problem we had for an automatic retrieval approach is that these values correspond exactly to the ones for mixtures between biomass burning aerosols and dust. It is therefore currently impossible to distinguish between a mixture of pollen and urban aerosols from a mixture of smoke and dust. With more features (for example, depolarization ratio or fluorescence capacity at other wavelengths...) such identification could become possible, and would therefore constitute a huge improvement from the current work.

2) Line 225. The authors claim that "the limited training set and excluding mixture cases and outliers from the repartition process." However, a significant number of outliers can still be observed in Figure 10b, for urban and smoke layers. This figure is not discussed in sufficient detail, and the presence of these outliers raises questions about the robustness of the classification model. More discussion of Figure 10b is necessary to clarify these issues.

Sentence line 225 concerns the choice of a threshold on the likelihood function for the identification of an aerosol layer from its depolarization ratio and fluorescence capacity.

If this threshold is too high, then, the larger circles in figure 4 are the ones used for the identification of the aerosol layers. This can create problems since with large clusters, we take the risk of having mixtures and outliers wrongly identified as belonging to the cluster of a specific aerosol layer. Making it difficult to exploit FLARE-GMM outputs.

If, on the opposite, the threshold is too low, then the smallest circles in figure 4 are considered for the identification of the aerosol layers. This is also problematic since the training set is limited. Therefore, aerosol layers having slightly different depolarization ratios and fluorescence capacity than the ones seen in the training set would not be identified by the algorithm.

Concerning the distribution represented in figure 10b. It corresponds to the distributions of altitudes at which the different aerosol types are identified in the Lille atmosphere. The outliers here concern therefore only the altitudes at which the aerosol layers are present. The outliers mentioned in line 225 concern outliers in terms of depolarization and fluorescence capacity, which we want to avoid having FLARE-GMM identifying them as either dust, smoke or urban aerosols. The presence of outliers figure 10b does therefore not question the robustness of the classification model.

3) Figure 11a: What is the reason of the high number of smoke layers observed in 2023? Please cite relevant studies.

2023 has been one of the years with the most fires in Europe and in Canada (Byrne et al. 2024, Kirchmeier-Young et al., 2024, San-Miguel-Ayanz et al., 2024...) explaining the high number of identified smoke cases in 2023, even though the fire activity in France was not as important as the one in 2022

(https://ec.europa.eu/newsroom/eusciencehubnews/items/842475/en?utm\_source=chatgpt.com).

However, figure 11 shows the number of lidar profiles which identify layers of smoke, urban, and dust for the different periods. This number varies with both the number of aerosol layers in the Lille atmosphere during the considered period, but also the availability of lidar measurements during this period. And according to the "Bulletin national de situation hydrologique de juin 2023" (<a href="https://www.eaufrance.fr/publications/bsh/2023-06?utm\_source=chatgpt.com">https://www.eaufrance.fr/publications/bsh/2023-06?utm\_source=chatgpt.com</a>) during May and June 2023, the anticyclonic conditions made that extremely few clouds and few precipitations have been registered in the North of France. It has therefore been possible to produce more measures with the lidar, and in higher altitudes, also potentially explaining the high number of identified smoke layers during this period.

4) Line 271. "The reason is that lidar can measure fluorescence and depolarization at very high altitudes, and in clear sky conditions, the *RH* level being typically lower at these altitudes." The authors must provide further justification or references supporting the assumption that RH is consistently low above 6 km in the regions and seasons studied.

Article form Wolf et al. (2023) has been cited in the revised version of the article.

5) Line 363. Do the authors have a reference to support their statement: "This rate is acceptable given that the two models use different features for classification and differ in their algorithmic structure" and the reason that this agreement rate can be considered acceptable for two classification schemes to provide the predominant type in the same atmospheric scheme?

To our knowledge, no study comparing the results of two different classification schemes to provide aerosol type has been conducted. The term "acceptable" may therefore not be adequate, and has been replaced by "encouraging" in the revised version of the paper.

6) Line 371. The inconsistency observed in the identification of desert dust is quite surprising. i) Have the authors examined some cases of inconsistence using other techniques or reference datasets to assess which of the two algorithms shows better agreement with the independent observations? ii) And, for the two dust specific events mentioned, what NATALI outputs were? iii) Could the presence of cirrus clouds, with high depolarization and low fluorescence, be another reason of disagreement between the two algorithms? iv) If so, have the authors checked the potential impact of the cirrus contamination on the classification results?3

In some occurrences, like the one presented, the results of FLARE-GMM and NATALI have been investigated to determine which of the two models was right. The case presented in Figure 9 corresponds to 21 March 2022. In this period, strong dust events can be observed in Lille (as represented in Figure 5 and Figure A2). Moreover, according to the Copernicus Atmosphere Monitoring Service (2021): CAMS global atmospheric composition forecasts (Copernicus Atmosphere Monitoring Service (CAMS) Atmosphere Data Store, DOI: 10.24381/04a0b097, Accessed on 29-Aug-2025). Desert dust aerosols appeared to be present in the atmosphere above Lille, reinforcing our confidence in FLARE-GMM retrieval.

Unfortunately, on the two dust events mentioned in part 4.1, we have not been able to estimate the aerosol type using NATALI. We have tried to contact the team in charge of the development of NATALI to have more information in that matter, but we did not receive any answer.

Eventually, concerning the presence of cirrus, it is true that it can affect both models' performances negatively. This is mentioned in part 5 concerning the retrieval of FLARE-GMM but also impacts NATALI since the identification of dust aerosols from cirrus clouds can be complicated using these lidar measurements. Unfortunately we did not have access to another instrument which would have allowed us to perform this identification. It is therefore complicated in this situation to estimate precisely the impact of cirrus on FLARE-GMM and NATALI performances. Further studies should be dedicated to that matter.

7) Figure 9: In Figure 9, the *Gfluo* signal appears quite noisy. Have any filtering or smoothing techniques been applied to the data shown?

No smoothing or filtering has been applied to  $G_{fluo}$ , instrument noise highly varies with atmospheric conditions.

**Typos and requests for clarification:**

8) Line 222: "It follows is that pollen aerosols are mainly located bellow the limit..." should be replaced with "As a result, pollen aerosols are mainly located below the detection limit..."

Done

9) Line 224: "making impossible to identify pure pollen aerosol layer", should be replaced with "making it impossible to identify a pure pollen aerosol layer..."

Done

10) Line 274: "allowing ut to perform aerosol typing up to 15 km," "ut" should be corrected to "us."

Done

11) Line 283: What the authors mean with "unseen cases". Please rewrite.

Unseen means that they are not in the training set, and that the machine learning model "hasn't seen them". It has been rewrite as "cases out of the training set".

12) Line 295: "strong manifestations of Sirocco winds have been experienced," should be replaced with "strong Sirocco winds were observed."

Done

**References:**

Byrne, B., Liu, J., Bowman, K.W. *et al.* Carbon emissions from the 2023 Canadian wildfires. *Nature* **633**, 835–839 (2024). <a href="https://doi.org/10.1038/s41586-024-07878-z">https://doi.org/10.1038/s41586-024-07878-z</a>

Kirchmeier-Young MC, Malinina E, Barber QE, Garcia Perdomo K, Curasi SR, Liang Y, Jain P, Gillett NP, Parisien MA, Cannon AJ, Lima AR, Arora VK, Boulanger Y, Melton JR, Van Vliet L, Zhang X. Human

driven climate change increased the likelihood of the 2023 record area burned in Canada. NPJ Clim Atmos Sci. 2024;7(1):316. doi: 10.1038/s41612-024-00841-9. Epub 2024 Dec 20. PMID: 39712870; PMCID: PMC11661968.

San-Miguel-Ayanz, J., Durrant, T., Boca, R., Maianti, P., Liberta`, G., Jacome Felix Oom, D., Branco, A., De Rigo, D., Suarez-Moreno, M., Ferrari, D., Roglia, E., Scionti, N., Broglia, M., Onida, M., Tistan, A. and Loffler, P., Forest Fires in Europe, Middle East and North Africa 2023, Publications Office of the European Union, Luxembourg, 2024, doi:10.2760/8027062, JRC139704.

Wolf, K., Bellouin, N., and Boucher, O.: Long-term upper-troposphere climatology of potential contrail occurrence over the Paris area derived from radiosonde observations, Atmos. Chem. Phys., 23, 287–309, https://doi.org/10.5194/acp-23-287-2023, 2023.

**Reply to RC2**

**Details:**

Abstract: All abbreviations need to be explained in the abstract.

Correction done.

Page 9, Figure 2: (b) needs y-axis text: PLDR as in (d).

PLDR has been removed from Figure 2 (d) instead to avoid overloading the figure.

Please add x-axis text in 2(a) and 2 (b): Fluorescence Capacity as in 2(c) and 2(d).

This choice was motivated by the intention to avoid overloading the figure, which already contains a lot of information, while ensuring that its overall readability is not significantly compromised.

**Section 4:**

Page 14, Figure 5: One should discuss the aerosol feature between 6-12 km height. The lofted layer is no longer visible when the optically dense cloud layer between 2-3 km height crossed the lidar. That should be mentioned as well

A comment has been added to clarify this.

Page 15, Figure 6: It seems to be that FLARE-GMM aerosol type estimation provides knowledge about the dominating aerosol type but not about the degree of mixing of different aerosol types. If that is true one should mention that.

Comments have been added in the abstract and in part 2.2 to clarify this.

Page 16, Figure 7(a): Please remove the bottom part of the figure, below the time scale.

Done

Page 17, line 345: The general performance includes what? Identification of all fractions of a given aerosol mixture?

Here, the performance of the algorithm corresponds to its ability to correctly identify the predominant aerosol type in a given layer. The fractions of a given aerosol mixture is an information

which is not possible to retrieve with FLARE-GMM so far. Further improvements could include the use of more features which could hopefully allow such retrieval.

Page 17, line 353: What is the difference between continental and continental polluted? Please explain!

Page 17, line 362: One should briefly summarize: What are the basic aerosol typing parameters in the case of NATALI (Angstroem exponent, color index, lidar ratio, what else?) and of FLARE-GMM (PDLR, G-Flour, RH).

The confusion matrix in Figure 8 is confusing: Dust is the most easy to identify aerosol type by using lidar. However, one has to make use of PLDR. If PLDR is not used in the aerosol typing scheme, the aerosol typing algorithm is simply not state of the art, and should not be used for comparison. One should at least mention this feature more clearly in the discussion. All modern lidars, including space lidars such as the CALIPSO lidar or the EarthCARE lidar, use the PLDR as one of the fundamental parameters in aerosol typing. The comparison is really limited if there are three aerosol types out of 5 that are very similar: continental, polluted continental, smoke.

Is there no progress in the case of NATALI? Do the NATALI developers still not make use of PLDR? What are the arguments of the NATALI developers for their selected strategy? One should contact them and discuss the results of the comparison with them, before the conclusions, based on the comparisons, are outlined. A more critical summary (as given on page 19, lines 391-394, is at the end needed). To repeat again, a modern aerosol typing scheme must include and make use of PLDR information! This is demanding! That should be clearly said.

Unfortunately, we have tried on numerous occasions to contact the team of Doina Nicolae at the National Institute of R&D for Optoelectronics, which has been in charge of the development of NATALI, but we were never given an answer. Unfortunately, our comprehension of NATALI is limited, and therefore, it is very difficult for us to answer these interesting questions that you raise.

Page 19, Figure 9: y-axix text is missing! Altitude [m]. And such a y-axis text plus numbers in m or km is needed in the right panel as well. Please also introduce (a) and (b).

(a) and (b) have been introduced, The y-axis text has been removed voluntarily from figure (b) as it the same axis as for figure (a)

Page 20, line 424: Cirrus PLDR is always >35% and dust PLDR < 30% at Lille at 532 nm, I would guess.

In our dataset, we have seen examples of desert dust layers exhibiting PLDR over 30% as shown in Figure 3, so it is very difficult to differentiate between cirrus and dust.

The statistics in Figure 10 include urban, smoke, and dust. Please discuss the potential impact of pollen (in the PBL). Most of the smoke is in the lowermost 2 km! I was expecting more smoke higher up (caused by all the plumes from North America reported in several Lille lidar papers).

Mixtures of urban and pollen are supposed to be attributed to the "unknown" class with FLARE-GMM as developed in section 3.2 considering that pollen and urban aerosols are in the same proportions. However, as the proportion of pollen decreases, FLARE-GMM should be able to correctly identify urban aerosols as being the predominant type. When investigating the dataset to identify pollen cases, we realised that this second scenario most often happens. The presence of pollen in the PBL is therefore not expected to have much impact on the results presented Figure 10.

Regarding the altitude at which biomass burning aerosols are identified, we were also surprised to see that a large amount of smoke layers are at low altitudes. However, we can still notice that the tail of the distribution goes much higher, which corresponds to the cases you mention.

Page 21, Figure 10: Please explain in detail, in the caption (preferably) or in the main text, what is presented: In (a), what show the error bars, what shows the blue area (without any numbers for the width). In (b), what shows the box, the horizontal line in the box, the error bars, the circles!

The violin plot (a) and the box plot (b) are representations very often used in data representation to quickly identify information on a distribution. Here, the distributions studied are the heights of aerosol layers for each aerosol type identified by FLARE-GMM. The shapes of the distributions are displayed on this figure and this is the classic form used for these kinds of plots (Tukey, 1977).

Page 22, Figure 11: ...function of time... Please explain the caption in (a) that1-3 bars are shown for ONE measurement event. Figure 11(a) is a bit confusing! Sometimes, I can see one bar, sometimes two, sometimes three bars. Sometimes the counts are large, sometimes very low. What show the counts exactly?

Figure 11 (a) shows three histograms. One for each aerosol type recognized by FLARE-GMM. The three histograms are superposed and while this form might be confusing, it is the best way to compare the three distributions of each aerosol type. Like the one displayed Figure 11 (b).

For each month of the dataset, the number of lidar profiles containing smoke, dust, and urban layers have been counted. The histogram shows these numbers for each month. When 1 bar only is present, it means that during the considered month, only one aerosol type has been identified.

**References:**

Tukey, John W.: Exploratory Data Analysis. Addison-Wesley Publishing Company Reading, Mass. — Menlo Park, Cal., London, Amsterdam, Don Mills, Ontario, Sydney 1977, XVI, 688 S., doi: https://doi.org/10.1002/bimj.4710230408